# Peptides Regulating Proliferative Activity and Inflammatory Pathways in the Monocyte/Macrophage THP-1 Cell Line

**DOI:** 10.3390/ijms23073607

**Published:** 2022-03-25

**Authors:** Francesco Avolio, Stefano Martinotti, Vladimir Kh. Khavinson, Jessica Elisabetta Esposito, Giulia Giambuzzi, Antonio Marino, Ekaterina Mironova, Riccardo Pulcini, Iole Robuffo, Giuseppina Bologna, Pasquale Simeone, Paola Lanuti, Simone Guarnieri, Svetlana Trofimova, Antonio Domenico Procopio, Elena Toniato

**Affiliations:** 1Department of Innovative Technology in Medicine and Odontoiatrics, Center of Advanced Studies and Technology University “G. d’Annunzio”, Chieti-Pescara, 66100 Chieti, Italy; avolio.francesco@gmail.com (F.A.); smartinotti@unich.it (S.M.); j.elisabetta.esposito@gmail.com (J.E.E.); giulia.giambuzzi@libero.it (G.G.); antonio.marino@asst-valcamonica.it (A.M.); riccardo.pulcini@unich.it (R.P.); 2Department of Biogerontology, Saint Petersburg Institute of Bioregulation and Gerontology, 197110 Saint Petersburg, Russia; vladimir@khavinson.ru (V.K.K.); katerina.mironova@gerontology.ru (E.M.); dr.s.trofimova@gmail.com (S.T.); 3Institute of Molecular Genetics, National Research Council, Section of Chieti, 66100 Chieti, Italy; robuffo@unich.it; 4Department of Medicine and Aging Sciences, Center of Advanced Studies and Technology University “G. d’Annunzio”, Chieti-Pescara, 66100 Chieti, Italy; giuseppina.bologna@hotmail.it (G.B.); simeone.pasquale@gmail.com (P.S.); paola.lanuti@unich.it (P.L.); 5Department of Neuroscience, Center of Advanced Studies and Technology, Imaging and Clinical Sciences, University of Chieti, 66100 Chieti, Italy; simone.guarnieri@unich.it; 6Department of Clinical and Molecular Sciences, Politecnic University of Marche, 60121 Ancona, Italy; procopioadg@gmail.com; 7INRCA-IRCCS, Clinic of Laboratory and Precision Medicine, 60121 Ancona, Italy; 8Unicamillus—Saint Unicamillus of Health Science, 00131 Rome, Italy

**Keywords:** peptides, LPS, monocytes, cytokines, STATs

## Abstract

This study evaluates the effects of five different peptides, the Epitalon^®^ tetrapeptide, the Vilon^®^ dipeptide, the Thymogen^®^ dipeptide, the Thymalin^®^ peptide complex, and the Chonluten^®^ tripeptide, as regulators of inflammatory and proliferative processes in the human monocytic THP-1, which is a human leukemia monocytic cell line capable of differentiating into macrophages by PMA in vitro. These peptides (Khavinson Peptides^®^), characterized by Prof. Khavinson from 1973 onwards, were initially isolated from animal tissues and found to be organ specific. We tested the capacity of the five peptides to influence cell cultures in vitro by incubating THP-1 cells with peptides at certain concentrations known for being effective on recipient cells in culture. We found that all five peptides can modulate key proliferative patterns, increasing tyrosine phosphorylation of mitogen-activated cytoplasmic kinases. In addition, the Chonluten tripeptide, derived from bronchial epithelial cells, inhibited in vitro tumor necrosis factor (TNF) production of monocytes exposed to pro-inflammatory bacterial lipopolysaccharide (LPS). The low TNF release by monocytes is linked to a documented mechanism of TNF tolerance, promoting attenuation of inflammatory action. Therefore, all peptides inhibited the expression of TNF and pro-inflammatory IL-6 cytokine stimulated by LPS on terminally differentiated THP-1 cells. Lastly, by incubating the THP1 cells, treated with the peptides, on a layer of activated endothelial cells (HUVECs activated by LPS), we observed a reduction in cell adhesion, a typical pro-inflammatory mechanism. Overall, the results suggest that the Khavinson Peptides^®^ cooperate as natural inducers of TNF tolerance in monocyte, and act on macrophages as anti-inflammatory molecules during inflammatory and microbial-mediated activity.

## 1. Introduction

Innate immunity represents a finely controlled network of physiological processes that synchronize immunocompetent cells to control and patrol the integrity of organs when exposed to injuries that may alter their functions [1,2,3,4]. Leukocytes are a qualified group of immune cells that are mostly involved in recognizing foreign organisms and in developing proper phagocytic activities [5,6,7]. In particular, monocytes are circulating leukocytes that move to sites of infection attracted by cytokines and chemokines, a specialized set of molecules that cooperate to induce terminal differentiation of monocytes into macrophages with intense phagocytic activity [8,9,10,11]. These large phagocytes are found in all tissues [12], where they patrol for potential pathogens through amoeboid movement. Besides phagocytosis, they play a critical role in nonspecific defense (innate immunity) and help in activating adaptive immunity functions by recruiting other immune cells, such as lymphocytes, by presenting antigens to T cells. In humans, dysfunctional macrophages cause severe diseases, such as chronic granulomatous disease, which result in frequent infections [13].

In addition to increasing the inflammation and stimulation of the immune system, macrophages play an important anti-inflammatory role, and can decrease immune reactions through cytokines release. Macrophages which promote inflammation are called M1 macrophages, while those that reduce inflammation and encourage tissue repair are called M2 macrophages [14]. This difference is reflected in their metabolism: M1 macrophages have the unique ability to metabolize arginine to the “killer” molecule nitric oxide, whereas M2 macrophages have the unique ability to metabolize arginine to the “repair” molecule ornithine [15,16].

Monocyte-derived cell lines with varying degrees of differentiation are often used to describe macrophage function, since they show clear advantages for ease of acquisition over primary macrophage in vitro cultures. In particular, the monocytic leukemic THP-1 cell line represents a suitable model for studying Khavinson Peptides^®^ effects on cellular proliferation and inflammatory activity [17].

These peptides propose a new challenge for understanding how natural and partially digested proteins can generate signaling molecules participating in the regulation of cell physiology. In 1983, V. Morozov and V. Khavinson demonstrated that cells produce low molecular weight peptide compounds, and that these peptides can mediate information transfer by regulating proliferation, differentiation and intercellular interactions [18,19]. These substances were discovered and identified in various tissues and called short peptides [20,21]. The isolated substances were all short-chain peptides with low molecular weight of up to 10,000 Da. During the last forty years, over twenty complexes of physiologically active peptides were extracted and characterized. Peptides were found to participate in gene-expression regulation and protein synthesis, according to the peptide cascade model, modulating multiple physiological functions of the organism [22,23,24]. These processes are aimed at preventing DNA damage, or its eventual suppression, and stimulating its repair, with the aim of restoring cellular homeostasis. Their main function is a normalizing effect on specific organs, replacing or supporting the activity of substances secreted by these morphological structures. They also increase the lifespan of mesenchymal stem cell (MCS) and induce molecular mechanisms that contrast senescence [25]. They modulate neuronal activity and hinder cognitive degeneration, probably by improving the integration of cellular functions, and thus counteracting the accumulation of free radicals. Furthermore, they implement the action of telomerase by promoting anti-aging effects that characterize many cell types [26]. Most of these peptides perform their function specifically on the organs from which they have been isolated. Several studies report that thymus involution is responsible for promoting harm in the elderly due to the reduction of immune reactions and the development of immune dysfunctions [27,28]. According to the reports of V. Khavinson, this observation set the stage for the use of therapeutic compounds with natural or synthetic thymic factors as immune modulators. Those natural thymic factors (NTFs) have been isolated from calf thymus by mild acid extraction (V. Morozov and V. Khavinson, 1991) and pharmaceutical products containing NTF (Thymalin) are used in clinical practice for the prevention and treatment of immunodeficiency [29,30]. Pharmaceuticals containing one of the immunomodulatory molecules (L-Glu-L-Trp) was derived from *Thymalin* by reversed-phase high performance liquid chromatography (RP-HPLC), termed *Thymogen* [31].

A large-scale study was conducted to isolate biologically active peptides from different human tissues (brain, blood vessels, heart, liver, bronchi, kidneys, pancreas and prostate, testes and others), and most of the peptides were synthesized in vitro and available for clinical usage. The administration of these complex peptide preparations on animals in various experiments restored and boosted the stage of resistance. Their chemical structure was established, and the synthesis from amino acids was carried out. The dipeptide Thymogen was obtained from a natural peptide complex, Thymalin, and its main biological target is the immune system. The tetra peptide Epitalon was obtained from the peptide complex Epitalamin, derived from a bovine pineal gland, and it is involved mostly in the neuroendocrine system [32,33,34]. The tripeptide Chonluten, a synthetic bronchial bio-regulator, involves the respiratory system as the principal biological targets [35,36]. The synthetic dipeptide Vilon is implicated in regeneration of eye retinal cells and brain neurons and promotes cell proliferation and wound healing [37,38,39].

Even though some of the immunomodulatory functions have already been investigated, the molecular structure of the peptide signaling is still not completely understood. Therefore, in order to characterize molecular signaling in detail and better understand the impact on cellular functioning, we used the following Khavinson Peptides^®^: Epitalon, Vilon, Thymogen, Thymalin, and Chonluten (hereinafter referred to as P1, P2, P3, P4, P5, respectively), as possible inducers of biological functions on THP1 cells used as an immune system cellular model. Monocytes can differentiate in vitro into macrophages after Phorbol 12-myristate 13-acetate (PMA) induction, maintaining the proliferative program in vitro. It was shown that peptides, in particular those derived from thymic and lymphoid organs, modulate the inflammatory activity of differentiated macrophages and show a progressive ability to increase the proliferative induction of monocytic cells in culture, synchronously influencing the various phases of the cell cycle [40,41]. It was shown that the increase in the proliferative rate is in part related to increased phosphorylation of (ERK1/2), as well as p70S6K [42,43], a latent cytoplasmic kinase activated by mitogenic stimulation. In addition, they also upregulate the phosphorylation of the c-Jun N-terminal kinase (JNK), functioning as a promoter of cellular detoxification once the amount of cell oxidation increases. In addition, when monocytic cells are forced, in vitro, to differentiate into macrophages, most of the peptides show a remarkable capacity to attenuate the expression of pro-inflammatory cytokines with a cytoplasmic signal transduction mechanism that is independent from a cytokine/receptor-mediated induction. Furthermore, in this study we show that the signal transducer and activator of transcription molecules, namely STAT1, are actively phosphorylated upon treatment with peptides and move towards the nucleus by acting independently of a ligand/receptor interaction.

Lastly, we observed, through cell–cell adhesion assay between LPS-activated human endothelial cells and the THP-1 monocytic cells, an effect of downregulation of adhesion activity.

These observations show that peptides may represent master regulators capable of influencing innate immunity and act as modulatory molecules during the inflammatory process.

## 2. Results

### 2.1. Phenotypic Markers of Cell Proliferation

The phenotype of macrophages derived from THP-1, an established cell line derived from a human monocytic leukemia, was analyzed. THP-1 cells can differentiate into macrophages after PMA treatment. PMA-treated THP-1 cells grew while adhered to a monolayer and showed cytological signs of macrophage differentiation. The cytofluorimetric analysis on differentiated cells showed (Appendix A) an increase in the cytoplasmic granulocytic density with a modulation of the cell shape and distinctive signs of cell differentiation [44].

THP-1 monocytic cells retain the capacity to grow in suspension and behave as a model of cell proliferation that could be influenced by several factors or culture conditions. The effect of peptides on THP-1 cell growth was analyzed incubating monocytes with standardized concentrations of these bio-regulators. By using a conventional growth rate assay, we measured the level of proliferation. Appendix A shows an increase in cell growth rate highlighted in all cell clones treated for each single peptide. Furthermore, FACS analysis was used to monitor all phases of the cell cycle and an equal distribution of cell percentage crossing all phases during cell cycling was observed, even though a moderate but consistent percentage of THP-1 showed an apoptotic profile when incubated with the bronchogenic derived Chonluten (P5) peptide (Appendix A).

Then, since extracellular vesicles act as shuttles capable of carrying nucleic acid molecules (DNA and RNA) or proteins out of the cell, they represent inflammatory and proliferative cell activity. Remarkably, it was observed that the level of EV production from THP-1 monocytic cells in active proliferation is influenced by the presence of peptides, as shown on Figure 1A, with a greater effect on Thymalin (P4) treatment, when co-incubated with LPS. Differentiated macrophages were also positively influenced by peptides in increasing the production of extracellular vesicle particles, although with slight variability between the different peptides analyzed. Surprisingly, Thymalin (P4) affects the production of EV much more. This might be related to the nature of the peptide product itself since Thymalin (P4) is a natural nonpeptide extract from thymus and may be influencing a higher EV release.

### 2.2. Induction of the Mitogen-Activated Phospho-ERK1/2 and Phospho-JNK in Cultured Monocytes

Since peptides have been described as inducers of anti-senescence mechanisms in stem cells [45]. The possibility that they can influence the proliferation of THP-1 cells in culture was investigated. In particular, the hypothesis that they can induce phosphorylation of cytoplasmic kinase sets, which are activated at various levels during mitogenic stimuli, was tested. As shown in Figure 2, cell extracts from THP-1 cells, treated with standard concentrations of peptides, show an increased phosphorylation of ERK1/2, a MAP kinase activated during the mitogenic process. All the peptides, except for Vilon (P2), can increase the phosphorylated level of ERK1/2, suggesting that they could influence the transcriptional or translational regulation of important cellular signaling molecules. The dipeptide Vilon (P2), on the other hand, shows a small modulatory capacity.

When THP-1 cells were incubated with the bacterial-derived lipopolysaccharide (LPS), which is an inducer of pro-inflammatory pathways, either alone or in the presence of the peptides, an addictive effect on tyrosine phosphorylation of ERK½ was only observed on extracts co-treated with Epitalon (P1) or Vilon (P2) (Figure 2A). No significant coordinated action between LPS and peptides influenced the activation of other kinases as the mTOR dependent p70 S6 (Figure 2C). Interestingly, a similar pattern of increased tyrosine phosphorylation was observed when specific anti-phospho antibodies raised against the stress related c-Jun N-terminal kinase (JNK) were used in a Western Blot assay of extracts derived from the peptide-treated THP-1 monocytes (Figure 2B). Surprisingly the co-incubation downregulated the activity of JNK with a reproducible exception of Thymalin (P4). This would suggest that when peptides are used as co-stimulators with pro-inflammatory inducers, a complex network of synchronized signaling events take place inside the cells.

### 2.3. Chonluten (P5) Promotes Mild Level of TNF Secretion in Monocytes

It was important to study the possible effects of peptides on cytokine release. Monocytes are usually refractory to cytokine influence when no inflammatory action is present. This is in part due to low expression of toll-like receptor 4 (TLR4) and cytokine-mediated signaling [44]. However, a slight moderate expression of TNF is documented on monocytes as a sign of self-tolerance to sustain a latent anti-inflammatory status during immune surveillance. Moreover, exposing THP-1 monocytes to different peptides in the presence or absence of LPS as an inducer of inflammatory pathways, and measuring cytokine release with cytometry analysis, no significant expression was evident on proliferating monocytes after LPS or peptide incubation. However, when monocytes were incubated with the Chonluten tripeptide (P5) (Appendix A), a slight release of the TNF cytokine was observed. This data suggests the possibility that such a bio-regulator acts in a transduction signaling mechanism that attenuates the inflammatory thrust by providing a type of immunological anergy [46].

### 2.4. STAT1 and STAT3 Are Directly Phosphorylated by Peptides on Differentiated Macrophages

In order to better investigate the molecular mechanisms that could influence the anti-inflammatory abilities of bio-regulators, Western Blot analyses were conducted to evaluate the activation status of the signal transducers and activators of transcription, namely STATs. STATs are a series of cytoplasmic signaling molecules mainly involved in cytokine activation, proliferative induction signals and inflammatory response modulation. As shown in Figure 3B, the treatment of macrophages with the pro-inflammatory inducer LPS activates STAT1 phosphorylation on a time course manner with maximum induction between four and eight hours after treatment. Co-incubation with standard amounts of peptides determines an increase in the phosphorylation process with the same temporal pattern, suggesting that the peptides may cooperate within the same transduction axis, or through an independent mechanism. This experiment confirms that at least Epitalon (P1), Vilon (P2) and the bronchial peptide Chonluten (P5), activate STAT1 phosphorylation, functioning possibly via a receptor-independent mechanism, not involving IFN-α/IFN-R interactions, as documented by no effect on IFN-α production (Figure 3A). As a matter of fact, several evidences suggest that the chemical structure of such peptides allows direct penetration into the cellular compartment regulating gene expression [22].

The possibility that peptides could regulate other STAT molecules involved in cytokine-mediated signaling, specifically those acting in the acute phase of inflammation, was also investigated. We tested several transducers of the STAT family, in particular the activator of transcription STAT3. STAT3 is mostly involved in IFN-α and IFN-γ mediated signaling, but it is also a major player in the IL-6 production and IL-6 mediated activity. Surprisingly, co-incubation of LPS and peptides do not determine any additive or synergistic effect on STAT3 phosphorylation. In particular the level of phosphorylated STAT3 slightly decrease and attenuates the activation mediated by the pro-inflammatory LPS (Figure 4). Moreover, the incubation of the THP-1 macrophages with the P1 peptide does not activate the phosphorylation status of the latent cytoplasmic transducers, since the anti-phospho-STAT3 antibodies did not show any significant increase in activated molecules. In addition, the P1 time–course treatment seems to further downregulate the phosphorylation level of phospho-STAT3 as compared to the control (Appendix A).

### 2.5. Cytokine Release Is Influenced by the Presence of Peptides in LPS-Treated THP-1 Cells

In order to study the pro-inflammatory cytokines releasing capacity of LPS activated macrophage cells while absence or while presence of peptides, we performed a flow cytometric analysis to detecting most of the pro-inflammatory and anti-inflammatory cytokine molecules. Despite some cytokines not being detected because of the limit of quantification (LOQ) due to sensitivity, it was shown that TNF-α, IL-6, and IL-17 at least, were reduced on cells co-incubated with peptides and LPS (Figure 5). IL-6 and TNF releases were almost downregulated by all the peptides in LPS media, whereas Thymogen (P3) was the most effective peptide attenuating IL-17 release from activated macrophages. Although we have not yet precisely identified the exact molecular mechanism that influences this inflammatory state, it is strikingly evident that the main pro-inflammatory signaling axis is modulated by the activity of peptides. This finding deserves a thorough investigation into probable peptide-induced gene regulation.

### 2.6. Confocal Microscopy Confirms That Peptide Mediates the Cytoplasmic STAT1 Transfer into Cells Nuclei

A clear shifting of the latent cytoplasmic STAT1 molecules into the nuclei of peptide-treated macrophages was observed through confocal microscopy. As evident in Figure 6, the control differentiated THP-1 cells did not show any significant transfer of the STAT1 into the nuclei. Whereas, upon peptide incubation, most of the peptides were able to activate cytoplasmic transducers passage into the nuclei as marked in Figure 6. The phosphorylation of the STAT1 molecules allows the unmasking of the nuclear localization site (NLS), determining the translocation into the cell nucleus. The conventional DAPI staining, that specifically binds strongly to adenine–thymine-rich regions in DNA, was used. Subtraction of the background staining was from incubation of fluorescence secondary specific antibody.

### 2.7. Cell–Cell Adhesion Assay between LPS-Activated Human Endothelial Cells and the THP-1 Monocytic Cells. Effect of Pretreatment with Bio-Peptides and Downregulation of Adhesion Activity

Human Umbilical Vein Endothelial Cells (HUVECs) were treated HUVEC with lipopolysaccharide (LPS) overnight to test adhesion molecule activation. ICAM-1 activation was determined using a Western Blot assay. The HUVECs were seeded on a specific photo-compatible multiple well plastic device and incubated with LPS for at least 12 h. Meanwhile, THP-1 monocytes were pretreated with specific bio-peptides for 8 h and labeled with DIL, before being transferred onto the monolayer of activated HUVEC cells for adhesion assay. Then, after removing the unbound cells, the monocytes adhering to the activated endothelial cells were quantified by counting the residual fluorescence and plotting the percentage of untreated monocytes adherent on the LPS-activated endothelial control cells versus the bio-peptide-treated monocytes. For each sample, ten fields were randomly selected at 20× optical zoom. For each field, all the nuclei stained with the DAPI and all the cells labeled with the DIL were counted. As can be seen in Figure 7A, the monocytes attached to the activated endothelial cells were recognized by the overlapping fluorescences (DAPI + DIL). As shown in Figure 7C, a significant decrease in the percentage of adherent monocytes occurred with respect to LPS-treated control HUVECs. All peptides were able to downregulate the adhesion mechanism of cell–cell interaction with the exception of Thymalin (P4) that conversely appeared to induce an increase in the interaction effect (Figure 7).

## 3. Discussion

In this study, we have investigated the proliferative and anti-inflammatory activity of Khavinson Peptides^®^ on the THP-1 human monocytic cell line, whose monocytes can switch phenotype differentiating into macrophages upon incubation with PMA. Monocytes are blood-circulating leukocytes acting as a primary line of defense in various areas of the body. Using THP-1 cells as an immune system cellular model, both mechanisms of replicative induction and of modulation of the inflammatory response were evaluated in relation to peptide influence on these functions at molecular level [47,48]. Since it has been shown these nano peptides may act as anti-senescence factors able to prolong the life span of mesenchymal stem cells in culture [33], monocytes were treated with a standard dose of peptides to investigate some cellular functions.

Initially, treating THP-1 monocytic cultures with specific peptides, we reported an increase in the growth and cell proliferation with a balanced distribution along the various cell cycle phases [49], and only the bronchogenic peptide Chonluten (P5) appears to increase the treated cells apoptosis level, doubling the value in relation to other peptides (Appendix A). Underlying this observation may be a unique behavior of Chonluten (P5) in the specific regulation of monocytic function through increased proliferation and, at the same time, modulation of apoptotic activity. [50]. As a matter of fact, since monocyte recruitment is quite active on bronchial and alveolar compartments during the extension of inflammation, the peptide itself could massively modulate the release activity of extracellular vesicles as a sign of cell viability associated with the proliferation index. These observations suggest that, under cell duplication conditions, peptides influence the duplication rate without causing dysfunction in the cell cycle phases.

To better understand the degree of monocytes proliferative induction, the level of activation of the MAP kinase complex depending on the peptide treatment was studied. As widely demonstrated, MAP kinases are key elements in the phosphorylation and dephosphorylation processes, acting in a critical interplay in macrophage biology [51]. In particular, the ERK1/2, which represents the goal of the mitogenic signaling inside the cell, showed various degrees of phosphorylation, possibly modulating the transcription of target genes. Indeed, as largely evidenced in recent years, extracellular signal-regulated kinases (ERKs) or classic MAP kinases are expressed protein kinases, which are involved in signal transduction mechanisms including the regulation of meiosis, mitosis and post-mitotic phases in differentiated cells [52]. Many different stimuli can activate the ERK pathway, including growth factors, cytokines, virus infection, ligands for heterotrimeric G protein-coupled receptors, transforming agents, and carcinogens. Therefore, it is reported here that both ERK1/2 phosphorylation on peptide-treated cells and specific activation of downstream signaling lead to the promotion of protein synthesis, as documented by the specific induction of the mTOR-dependent p70S6 kinase.

It is important to underline how the in vitro peptide treatment of proliferating cells affects the phosphorylation level of kinases, such as ERK1/2, involved in the mitogenesis process. Sure enough, this signal axis is important in mediating the proliferative effect through the increase in glucose-dependent metabolism, as it is widely demonstrated in mTOR-dependent signaling.

The activation of the c-Jun N-terminal JNK kinase by peptide regulators also seems to play an important role [53]. JNK represents a class of cytoplasmic kinases involved in signal transduction pathways activated during stress and increased exposure to oxidant species. Since several reports have stated that most peptides synchronously regulate and sustain antioxidant activity, it is quite interesting to observe that the same molecules promote JNK activation, suggesting a possible mechanism related to detoxification programs which are active on monocytes during intense cell proliferation. It is of interest to highlight (Figure 2) a significant activation of the p70-S6 kinase, whose coordinated signaling is downstream with respect to the activated molecules induced by mitogenic factors. The p70-S6 kinase is activated by mTOR-dependent transduction mechanisms to satisfy the greater glucose-dependent energy demand seen during cell proliferation. JNK activation also reflects this mechanism, with relevance to stress induced activity, such as the ability to counteract the negative effects of increased oxidation and free radical induced cell damage already seen in other cell types.

Whereas it would be very important to analytically investigate molecular mechanisms of gene activation to better understand the action of peptides in vivo, it is here reported how most of the investigated Khavinson Peptides are able to activate the phosphorylation of STAT molecules, specifically STAT1, acting alone or in co-treatment with LPS on macrophage cells. STAT1 belongs to the class of signal transduction and transcription molecules that cooperate with receptor associated kinases in transferring the cytokine-mediated biological response into the nuclei [54]. This involves an activation that is time dependent and reflects a mechanism of fine transduction regulation that takes place within the treated cells. Furthermore, it is important to point out that the STATs seem to be modulated by peptides in a receptor-independent mechanism, as stated by the observation that IFN-α modulation is not affected by amino acids or Epitalon (P1) treatment (Figure 3A). Considering that STAT1 reflects more properly a mechanism linked to the induction of type I Interferon involved in the control of viral infection through the binding to the IFN-α type one receptor, it could be plausible to speculate about the role of these peptides to mimic a similar effect.

It should also be noted that the peptides seem not to induce STAT3 transducer activation, but rather showed a downregulation capacity on the STAT3 phosphorylation, albeit not too marked, in the presence of the pro-inflammatory activator LPS. Therefore, these documented observations suggest that these peptides could downregulate pro-inflammation on activated macrophages, by decreasing of STAT3 phosphorylation. As a matter of fact, STAT3 is the most involved transducer activated by acute inflammatory stimuli that increase the transcription of IL-6, one of the main players in the acute phase response during inflammation and infectious diseases [55]. As evidenced in Figure 5, the cytokine levels of IL-6, TNF and to a lesser extent of the IL-17, decrease significantly on macrophages activated by LPS, after co-incubation with peptides.

Finally, the experiment of co-incubation of peptide-treated monocytes with LPS-activated human endothelial cells, freshly isolated from the umbilical cord, highlights that the peptides attenuate the adhesion mechanism that acts between endothelium and immune cells, as shown in Figure 7. Especially P3, (Thymogen), seems to drastically reduce the percentage of adherent treated monocytes, which approaches the percentage of the negative control (endothelium not activated with LPS). Although all the synthetic purified peptides have the aforementioned effect, the natural organ extracted Thymalin (P4) appears not to exhibit this ability probably due to its composition. These results support the hypothesis that, among different anti-inflammatory properties, most bio-peptides regulate the cellular interaction mechanism of the immune system, as documented in our vitro model.

This also would pre-suppose a clear anti-inflammatory effect within the functional mechanism evidenced by macrophages subject to intense inflammatory activity. What could be the value of Khavinson Peptides as modulators of this anti-inflammatory function, especially in relation to the IL-6 cytokine and to the capability of downregulating key adhesion molecules in immune cells, is a question widely open to scientific investigation.

## 4. Materials and Methods

### 4.1. Peptides

Five different Khavinson Peptide preparations were used: P1 as Epitalon (Ala-Glu-Asp-Gly, from pineal gland), P2 as Vilon (Lys-Glu, a synthetic immunomodulatory dipeptide), P3 as Thymogen (dipeptide Glu-Trp, from calf thymus), P4 as Thymalin (peptide complex, from calf thymus), P5 as Chonluten (tripeptide Glu-Asp-Gly, from respiratory lung). All peptides derived specifically from organ of animals and some (Epitalon, Vilon, Chonluten) were synthetized on the basis of the information of their natural counterparts.

### 4.2. Cell Culture

Human leukemia monocytic cell line (THP-1) was purchased from ATCC. Cells were grown in RPMI 1640 (Sigma–Aldrich; Merck Millipore, Darmstadt, Germany) supplemented with 10% FBS, 2 mM L-glutamine and 1% Pen–Strep (Thermo Fisher Scientific, Inc., Waltham, MA, USA), and were maintained at 37 °C in a humidified atmosphere of 5% CO_2_ in air. Two series of experiments were performed when the cells reached 80% confluence.

THP-1 monocytic cells (~2 × 10^5^ cells/mL) were plated as a control group *w*/*o* treatment or treated overnight with 100 ng/mL per peptide (P1–P5) alone or supplemented by 100 ng/mL lipopolysaccharide (LPS). LPS derived from *E. coli* was purchased from Sigma–Aldrich, Saint Louis, MO, USA. Subsequently monocytes were collected by centrifuging for 8 min at 1000 rpm and cell extracts were made according to standard procedures

Cell extracts and differentiation of THP-1 cells into macrophages were achieved using the methods described by Daigneault et al. [56]. THP-1 cells (~2 × 10^5^/mL) were incubated with 100ng/mL PMA (Sigma–Aldrich, Saint Louis, MO, USA) for 3 days at 37 °C, in order to induce monocyte–macrophage differentiation. The PMA-containing media was removed, and cells were incubated for a further four days in conventional medium, before treating THP-1-derived macrophages on a time–course experiment (2, 4, 8 and 12 h) with 100 ng/mL of every peptide alone or supplemented with 100 ng/mL LPS, according to a specific time–course.

THP-1-derived macrophages were detached incubating into Trypsin-EDTA 1X (Euro Clone) for 10 min at 37 °C. Monocytes and macrophages were counted using 0.4% of Trypan–blue solution (Sigma–Aldrich, Saint Louis, MO, USA) in a Burker chamber and tested for viability.

For Amino Acid treatment, a mixture of L-isoleucin, L-leucin, L-alanine and L-valine was used where indicated (Freliver Sport, Dompé Co., L’Aquila, Italy).

HUVECs were provided as a donation by the research group of Professor Mario Romano of the University of Chieti. These cells were isolated by collagenase-1 treatment on umbilical veins collected from women undergoing cesarean section in the department of Obstetrics and Gynecology, obtained from randomly selected healthy mothers delivering at the Chieti University Hospital (Chieti, Abruzzo, Italy), using 0.1% collagenase at 37 °C (with ethical clearance and written informed consent) [57]. Separated cells were grown in appropriate medium (DEM low glucose and supplemented with 10% FCS and 1% Pen–Strep).

### 4.3. Cell Viability

The cell viability was measured with 3-(4,5-dimethylthiazol-2-yl)-2,5-dipenyl tetrazolium bromide (MTT) (Sigma–Aldrich, Saint Louis, MO, USA) assay according to standard procedures for adherent cells. Absorbance (optical density) was measured at 530 nm using a micro-plate reader (Infinite F50, Tecan, Männedorf, Switzerland). The experiment was performed in triplicate.

### 4.4. Flow Cytometry and Cell Cycle Analysis

Cell cycle investigations were performed according to published protocols [58,59]. Cell samples from the T25 flasks were collected, washed with PBS (phosphate buffered saline) and centrifuged (400× *g*, 10 min at room temperature, RT). Cell pellets (5 × 10^5^ cells/sample) were fixed by adding 500 µL of 70% cold ethanol and then stored at 4 °C. After at least one week, cell pellets were washed, re-suspended and stained by adding 500 µL of a solution containing 50 µg/mL of propidium iodide (PI, Sigma–Aldrich, Milan, Italy) and 200 µg/mL of RNase A (Sigma–Aldrich). Samples were incubated overnight at 4 °C in the dark and then acquired by flow cytometry (FACS Canto II, BD Biosciences, San Jose, CA, USA). PI fluorescence data were collected using linear amplification. Debris was excluded from the analysis by gating on morphological parameters. A minimum of 10^4^ events were recorded for each sample. Each sample was prepared and analyzed in triplicate. Data were analyzed using FlowJo X v 10.0.7 (TreeStar, now Becton, Dickinson and Company, Ashland, OR, USA). Aggregates were excluded [58,59], and the percentages of cells in G0/G1, S, and G2/M phases of the cell cycle, as well as apoptotic and necrotic populations, were calculated.

In order to identify the activated compartment, THP-1 cells (500 × 10^3^/sample), after centrifugation at 400× *g* for 10 min, were washed in PBS and stained using the PE-conjugated anti-CD25 (BD Biosciences, San Jose, CA, USA). After a 30-min incubation at 4 °C with the antibody, the cells were washed, re-suspended in 0.5 mL PBS, and 100 × 10^3^ events/sample were acquired using a flow cytometer (FACS Canto II; BD Biosciences, San Jose, CA, USA). The gates of CD25+ cells were placed based on the relative unstained sample. A forward scatter area (FSC-A) vs. side scatter area (SSC-A) dot plot was used to identify and gate THP-1 cells. The cytometer setup and tracking module (CS&T) (BD Biosciences, San Jose, CA, USA) was used to test instrument performances and data reproducibility; these were further validated by the acquisition of rainbow beads (BD Biosciences, San Jose, CA, USA) and compensation was calculated using comp beads (BD Biosciences, Franklin Lakes, NJ, USA). Before each analysis, the flow cytometer was cleaned according to the manufacturer’s instructions, and carryover between samples was prevented using the FACSCanto II SIT Flush device. The FACSDiva v6.1.3 or FACSuite v 1.0.6.5230 (BD Biosciences, San Jose, CA, USA) were used to analyze the data after the acquisitions.

### 4.5. Flow Cytometry Extracellular Vesicle Staining, Acquisition and Analysis

Extracellular vesicles (EVs) represent extracellular shuttles capable of transporting nucleic acid molecules (DNA and RNA) or protein outside the cell. For the EV staining, a method recently optimized in our laboratories was applied [60,61]. In detail, EVs were stained by adding 0.5 µL of FITC-conjugated phalloidin and lipophilic cationic dye-LCD (BD Biosciences—Catalogue, #626267, Custom Kit), to 100 µL of supernatants. After 45 min of staining (RT, in the dark), 200 mL of PBS 1× were added to each tube and 1 × 10^6^ events/sample were acquired by flow cytometer (FACSVerse, BD Biosciences). Parameters were optimized as already described [60,61]. All the state-of-the-art recommendations were taken into consideration.

The evaluation of non-specific fluorescence was obtained by acquiring fluorescence minus one (FMO) and the isotype controls. Compensation was assessed using comp beads (BD Biosciences) and single stained fluorescent sample [62]. Data were analyzed using FACSuite v 1.0.6.5230 (BD Biosciences) and FlowJo X v 10.0.7 (Tree Star, Ashland, OR, USA) software. EV concentrations were obtained by volumetric count [60,61,63,64].

### 4.6. Protein Extraction and Western Blot

Total proteins were extracted from monocytes and macrophages in 50 mM Tris-HCl pH 7.8, 1% Triton X100, 0.1% SDS, 250 mM NaCl, 5 mM EDTA lysis buffer in the presence of the mini protease inhibitor cocktail at 150 µL/mL (Roche Diagnostics, Mannheim, Germany) as described [65]. Cell lysates were separated on 12% or 15% SDS-PAGE depending on molecular weight of target proteins and transferred to a polyvinylidene difluoride membrane (Bio-Rad, Hercules, CA, USA). Membranes were incubated overnight at 4 °C with specific primary Abs (diluted 1/100, Santa Cruz Biotechnology, Dallas, TX, USA). Antibodies directed against their protein target were purchased from Cell Signaling Technologies (Danvers, MA, USA). Membranes were incubated overnight at 4 °C on a shaker with specific primary antibodies to phospho-SAPK/JNK, phospho-p70 S6 kinase, phospho-p44/42 MAPK (ERK1/2), IFN alpha, and β-actin as a control. Specific secondary goat anti-rabbit IgG-HRP conjugated antibodies (1:2000, Santa Cruz) (dilution according to Cell Signaling Technology specific protocol) were used for detections in all experiments, incubating membranes for one hour at RT with gentle shaking. Immunocomplexes were visualized using the ECL detection system (Amersham Pharmacia Biotech, Little Chalfont, UK). Western Blot densitometry band quantification was by ImageJ software (Rasband, 1997–2014).

### 4.7. Confocal Microscopy

The STAT1 marker was analyzed by confocal microscopy detection (Zeiss LSM800, Zeiss, Jena, Germany) using anti-antibody and nuclear-ID red stain (Cell Signaling Technologies, Danvers, MA, USA). Cells were grown, treated and differentiated on coverslips according to standard protocols. Then, they were fixed with a 4% paraformaldehyde solution, permeabilized with 0.25% Triton X-100 (Sigma-Aldrich, St. Louis, MO, USA) and incubated with primary antibody (anti-STAT1, D4Y6Z, Cell Signaling, diluted 1:100 in 10% Goat Serum) and then immunocomplexes reacted with secondary antibody (Alexa Fluor 488 conjugated, diluted1:100 in 10% Goat Serum) (Invitrogen, Carlsbad, CA, USA). DAPI staining (4′6-diamidino-2-phenylindole) was used to visualize the nucleus (Invitrogen, Carlsbad, CA, USA). The capture settings were kept constant between different acquisition sessions.

### 4.8. Cytokine Analysis

Concentration of several cytokines, released during biopeptide treatment, was determined using flow cytometry (Becton Dickinson, NJ, USA) [65,66].

The release of cytokines (IL-2, IL-4, IL-6, IL-10, TNF, INF-γ, IL17-A) in supernatants was measured by flow cytometry using the commercial Cytometric Bead Array (CBA) Human Th1/Th2/Th17 Cytokine Kit (BD Bioscience catalogue 560484), following the manufacturer’s instructions. Samples acquisition was performed on FACSVerse (BD Biosciences) with FACSuite v 1.0.6.5230 (BD Biosciences) and data analysis by FCAP Array Software v3.0 (BD Bioscience) as suggested.

### 4.9. Monocytes Adhesion Assay

THP-1 monocytes were cultured at 1 × 10^6^ (5 × 10^5^ × cm^2^) and treated with Khavinson Peptides for 8 h. After treatment, the THP-1 cells were then washed with PBS (1x) and re-suspended in 300 μL of HUVEC medium [66] and labeled for 30 min with 10 μL of DIL stain (1,1′-Dioctadecyl-3,3,3′,3′-Tetramethylindocarbocyanine Perchlorate; cat# D282, Thermo Fisher Scientific). HUVECs were seeded at 2 × 10^4^ cell per cm^2^ in gelatin-pretreated (0.2%) single well dish (Ibidi treat 60 µ-Dish) for quantification assay. To activate the endothelial cells, treatment with LPS 5 µg/mL (from Escherichia Coli 026: b6, cat# L2654, Merck, NJ, USA) was performed for 12 h. Cells were then washed with PBS (3x) and co-incubated with 1 × 10^6^ THP-1 labeled monocytes added on the top of the endothelial monolayer at 37 °C for 30 min [67].

### 4.10. Validation Using Confocal Microscopy

After removing the unbound monocytes with PBS (3×), the HUVEC and bound monocytes were fixed in paraformaldehyde (2%) for 30′ and then wash in PBS (3×) After nuclear counterstain using DAPI cells were visualized using a ZEISS LSM 800 confocal laser scanning microscope [67].

### 4.11. Statistical Analysis

Data are represented with bar graphs as the mean ± SD of three independent experiments.

## Figures and Tables

**Figure 1 ijms-23-03607-f001:**
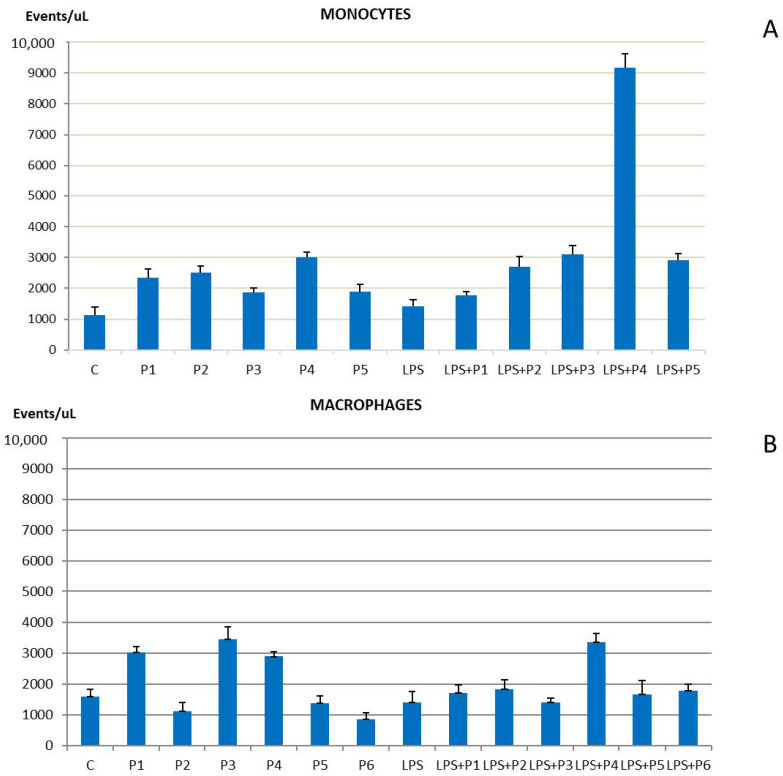
Extracellular vesiclescount through cytofluorimetric analysis on monocites (**A**) and macrophages (**B**). Extracellular vesicles represent shuttles capable of carrying nucleic acid molecules (DNA and RNA) or proteins out of the cell. Extracellular vesicles concentration generally reflects inflammatory or proliferative mechanisms and it is a function of mitogenic activity. Bar graphs show the mean values of three independent experiments.

**Figure 2 ijms-23-03607-f002:**
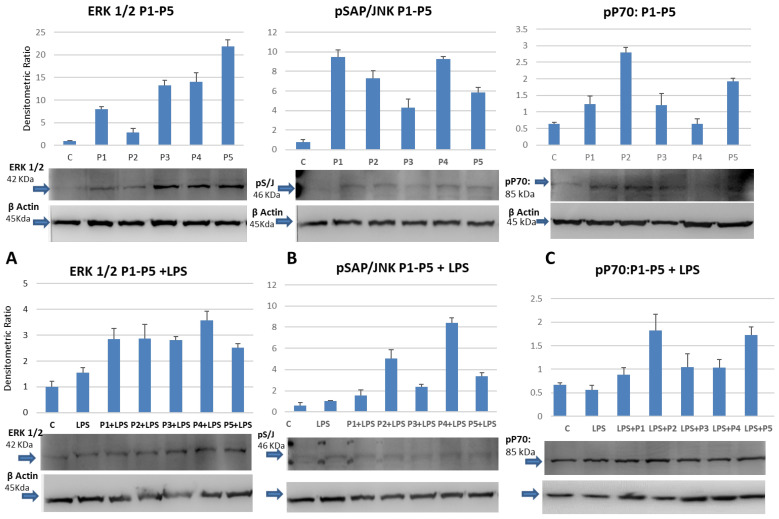
Induction of the mitogen-activated phosho-Erk1/2(4.(**A**)), SAP/JNK(4.(**B**)) and p70(4.(**C**)) (in established monocytes). P1, P2, P3, P4, P5 represent Epitalon, Vilon, Thymogen, Thymalin and Chonluten, respectively as indicated. Bar graphs show the mean values of three independent experiments.

**Figure 3 ijms-23-03607-f003:**
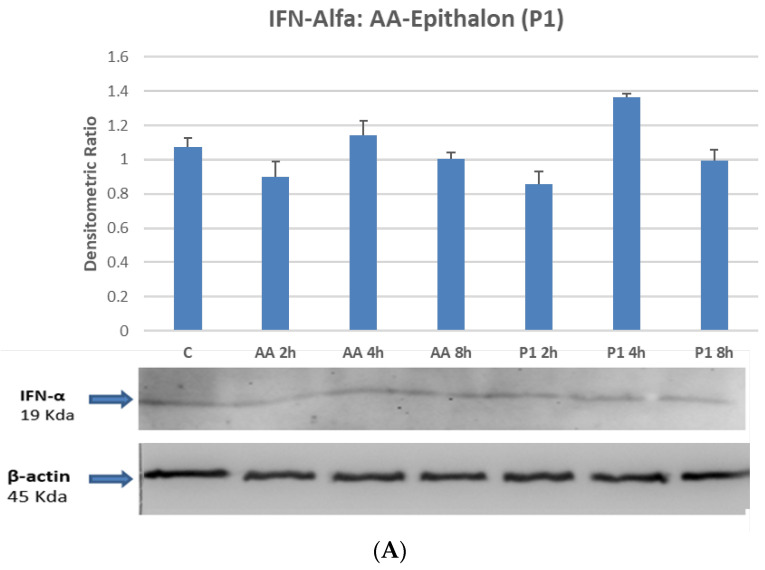
(**A**) IFN-a Detection of IFN-a by Western Blot assay. THP-1 cells were treated with either P1 peptide or a mix of amino acids (AA). No significant modulation was observed among the different pools of extracts. (**B**) Synergistic activation of STAT-1 by LPS. Epitalon (P1), Vilon (P2) and Chonluten (P5) biopeptide treatments of macrophage differentiated THP-1 cells. Phospho-activation of STAT-1 by bio-peptides on a time course treatment of differentiated macrophage THP-1 cells Bar graphs show the mean values of three independent experiments.

**Figure 4 ijms-23-03607-f004:**
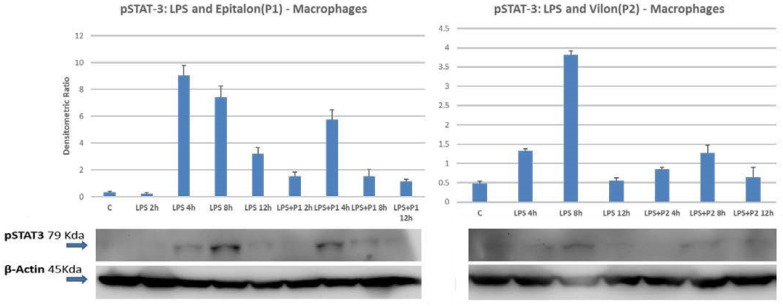
pSTAT3 activation upon LPS and LPS + bio-peptides. Densitometric analysis was performed normalizing band intensity to β-actin expression. Bar graphs show the mean values of three independent experiments.

**Figure 5 ijms-23-03607-f005:**
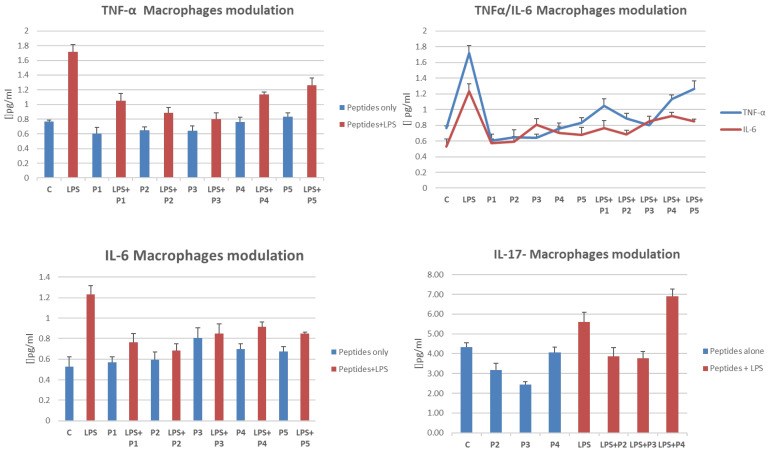
Macrophage cytokine modulation: TNFα, IL-6. and IL-17. Supernatants from treated cells or controls were used as described in Materials and Methods. Data are a mean of at least three independent experiments.

**Figure 6 ijms-23-03607-f006:**
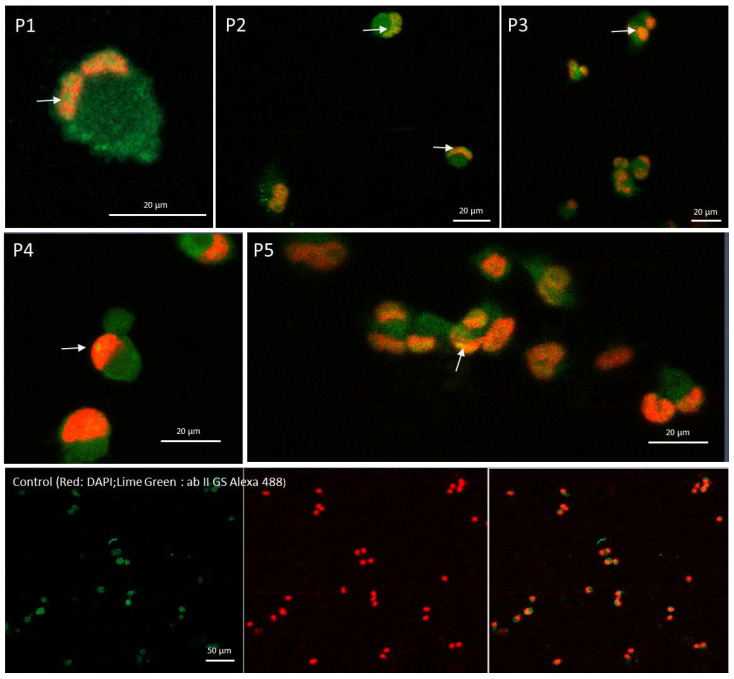
Confocal microscopy using fluorescent antibodies raised against STAT1. Nuclei were labeled with the DAPI staining. Red background was used for nucleus staining versus green-labeled antibody reacting with anti-STAT1. Yellow spots indicated by arrows show transfer of activated STAT1 from cytoplasm to the nucleus. At the bottom, the scale bar represents the micrometer reference of each microscopic field.

**Figure 7 ijms-23-03607-f007:**
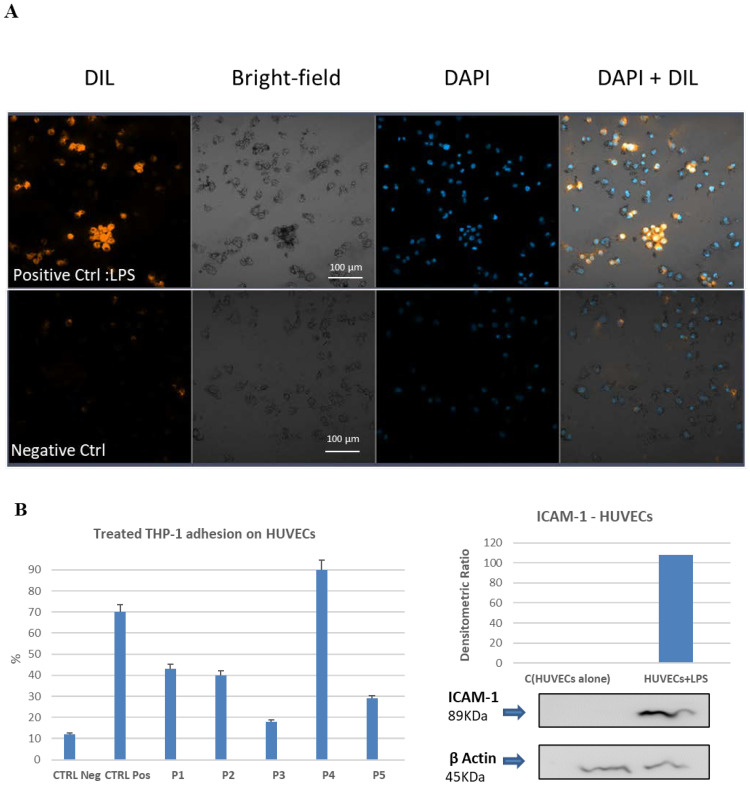
Cell–Cell Adhesion Assay: Confocal microscopy quantification and ICAM-1 Western Blot analysis. (**A**) First column shows fluorescence DIL, the second column shows bright field, the third fluorescence DAPI and the fourth shows fluorescence DAPI + DIL of the same field (20×). The negative control shows HUVECs alone (without LPS activation), the positive control shows THP1 monocyte adhesion to HUVECs. (DAPI: 4,6-diamidino-2-phenylindole; DIL: Dioctadecyl-3,3,3,3 -tetramethylindocarbocyanine perchlorate; HUVEC: Human umbilical vein endothelial cell; THP1: THPI monocyte). The scale bar expresses the micrometric reference. (**B**). On the left, bar graph represents the mean of three independent experiments on the effect of peptides on cell adhesion. On the right, Western Blot analysis shows the I-CAM1 induction by LPS-treated HUVECs as proof of adhesion activation. Bar graph shows the mean values of three independent experiments. (**C**) P1 to P5 represent experiment upon peptide treatment of THP-1 monocytes. On the right, percentage of adhesion is shown corresponding to each peptide utilized. On the left, confocal images of fluorescence relative to the adherent monocytes is depicted according to the scheme of Figure 7A, without the bright field images. The scale bar expresses the micrometric reference.

## Data Availability

Not applicable.

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
