# Peer review of "Peptides Regulating Proliferative Activity and Inflammatory Pathways in the Monocyte/Macrophage THP-1 Cell Line"

_ijms, 2022, doi:10.3390/ijms23073607_

Round 1

Reviewer 1 Report

Avolio et al. report on the influence of Khavinson peptides on inflammatory and proliferation processes that occur in monocytic cell lines. As a consequence, the peptides could promote monocyte differentiation into macrophages. The authors applied several distinct techniques to conduct their studies, and a respectable amount of work was done and described. However, in my opinion, the paper suffers major flaws precluding its publication:

  1. My significant doubt is formal and is related to ethical issues. The authors declare that the study was conducted according to precepts outlined in the Declaration of Helsinki. However, they did not provide any approval from an ethics committee (number and date of the approval should be provided). Additionally, the authors come from two countries. Was the permission obtained from the ethics committee from Italy, Russia, or both countries? In which country the HUVECs cells were isolated?
  2. Khavison peptides have been well known for several decades. The authors should explain the novelty of their study in the light of other results related to the biological influence of these peptides on human cells.

Minor issues:

  1. In some parts of the manuscript, the authors provide various styles, random italicized. The manuscript should be carefully checked prior to the resubmission.
  2. The authors mark individual peptides to P1 -P5 in the materials and methods section and use this system in figure captions. On the other hand, they usually use peptide names through the text. It should be unified.

Reviewer 2 Report

Review:

In the current manuscript, Francesco A et al. aims to characterize the biologically active peptides Epitalon, Vilon, Thymogen, Thymalin, and Chonluten in THP-1 and HUVEC cells in vitro. They studied for their modulating proliferative patterns, tyrosine phosphorylations of cytoplasmic kinases, and cytokine regulations with or without LPS. In the present study, the authors showed these peptides are natural inducers of TNF tolerance in monocyte and act on macrophages as anti-inflammatory molecules during inflammatory and microbial-mediated activity. The article is written in good detail, results are presented with experimental evidence. I would suggest authors improve the manuscript in considering the following major comments.

  1. The abbreviations must be kept uniform throughout the article. Example Line no:144 "1% Pen-Strep", Line: 280 "100 U penicillin, 100ug streptomycin/ml". 
  2. The figures are refereed thought articles, not in uniform. Example line no: 323 "Figure 1A " but Line:318 " (Fig. 2 B of supp. data)", Line 351:"(Fig. 2.A)". kept uniform throughout the article.
  3. The first time acronym used in the abstract and article must be spelled out Example: TNF, THP-1.
  4. In Result section line no: 324 "Differentiated macrophages were also positively influenced by peptides in increasing the production of extracellular vesicle particles although with slight variability between the different peptides analyzed' However, figure 1A shows LPS+ P4 is positively influenced other peptides do not have any significant influence. Could you explain this?
  5. In Figure 2A, Erk1/2 p1-p5+LPS, the Y-axis is missing parts. Further Figure 2C pP70:p1-p5, the pP70 western blot images are fuzzy, not visible in all bands, Could you improve the figure?
  6. I suggest moving Figure 3, to supplementary, since provides P5 only active TNF can be mentioned in the text easily.
  7. Line: 389, have "determines an increase in the phosphorylation process with the same temporal pattern,,". remove double commas.
  8. Inline no: 413, "Incubation of the THP-1 macrophages with the peptides as de as depicted alone do not activate phosphorylation status of the latent cytoplasmic transducers (data not shown)". It is advisable to show this data in supplementary figures.
  9. Figure 6, Modify the figure to fit in A4 size. After printing in the A4 sheet, the figure legends are missing parts. 
  10. In line:457, Some of the text is part of methods, remove or rewrite these results.
  11. In figure 8: The figures are not aligned properly mainly in figure 8C confocal images need to be aligned properly. In addition, consider removing the box and keeping the figures uniform style throughout the article.
  12. I have major comments in the discussion sections,  
  • Discussions are not refereed to the articles, throughout the discussion I can see only one reference mentioned.
  • This discussion section must be rewritten and properly discussed the authors finding with proper references.
  • In many places in the discussion section, authors claim claims statements without the reference or evidence in the article. Example line no: 509-5011 "In addition, by investigating the cellular distribution along with the phase of the cell cycle, only the bronchogenic peptide Chonluten appears to increase the level of apoptosis of the treated cells doubling the value in relation to other peptides." The level of apoptosis in different cell cycle phases not be seen throughout the article.

Round 2

Reviewer 1 Report

After reading the responses to my comments, I recommend accepting the manuscript for publication in the International Journal of Molecular Sciences.

Reviewer 2 Report

The author addressed most of the comments in the first review, still advisable to improve the discussion section. Further, I have the following comments that need to be addressed.

  1. Most of the experiments have not been performed in triplicate (Figure 1,2,3,4, and 7), since there is no information, Can you explain why?
  2. Figures 6 and 7, do not have a scale bar in the image, could you update the figure with a scale bar.

Round 3

Reviewer 2 Report

Thanks for the author addressing all my comments. I do not have any further comments.